# The Japanese Herbal Medicine Yokukansan Exerted Antioxidant and Analgesic Effects in an Experimental Rat Model of Hunner-Type Interstitial Cystitis

**DOI:** 10.3390/medicina58060810

**Published:** 2022-06-15

**Authors:** Tatsuki Inoue, Mana Tsukada, Yoshiki Tsunokawa, Yoshiko Maeda, Seiya Fukuoka, Takashi Fukagai, Yoshio Ogawa, Masataka Sunagawa

**Affiliations:** 1Department of Physiology, School of Medicine, Showa University, Tokyo 142-8555, Japan; tatsuki22@med.showa-u.ac.jp (T.I.); t.yoshiki@med.showa-u.ac.jp (Y.T.); seiya@med.showa-u.ac.jp (S.F.); suna@med.showa-u.ac.jp (M.S.); 2Department of Urology, School of Medicine, Showa University, Tokyo 142-8666, Japan; ymaeda@med.showa-u.ac.jp (Y.M.); fukagai@med.showa-u.ac.jp (T.F.); ogawayos@med.showa-u.ac.jp (Y.O.)

**Keywords:** Hunner-type interstitial cystitis, Yokukansan, antioxidant effect, hydroxyl radical, Kampo formula

## Abstract

*Background and Objectives*: The Japanese herbal medicine Yokukansan (YKS) has analgesic properties and is used for various pain disorders. The purpose of the present study was to investigate the effects of YKS in Hunner-type interstitial cystitis (HIC) using an experimental rat model of HIC and to explore its antioxidant activity and role as the underlying mechanism of action. *Materials and Methods*: The antioxidant capacity of YKS was evaluated by determining its hydroxyl radical (·OH) scavenging capacity using electron spin resonance (ESR). Next, the effects of YKS administration were explored using a toll-like receptor-7 agonist-induced rat model of HIC. The von Frey test was performed to assess bladder pain. Three days after HIC induction, the bladder was removed, and the expression of oxidative stress parameters in the bladder wall was investigated (reactive oxygen metabolites (ROMs), ·OH, and 8-hydroxy-2′-deoxyguanosine (8-OhdG)). *Results*: YKS had a ·OH scavenging capacity according to the ESR study. In the von Frey test, a significant decrease in the withdrawal threshold was observed in the HIC group compared with the control group; however, the decrease was ameliorated by the administration of YKS. Oxidative stress parameters showed increasing tendencies (ROMs test and 8-OHdG) or a significant increase (·OH) in the HIC group compared with the control group; however, the increase was significantly suppressed by the administration of YKS. *Conclusions*: These findings suggest that YKS is effective against HIC and that its antioxidant activity is involved in the mechanism of action.

## 1. Introduction

Interstitial cystitis/bladder pain syndrome (IC/BPS) is a nonspecific chronic inflammatory disease of the bladder that causes lower urinary tract symptoms, such as urinary frequency, urinary incontinence, voiding pain, and chronic pelvic pain [1,2]. As in most cases, its etiology is unknown; IC/BPS is considered an idiopathic disease [3]. It is divided into Hunner-type (HIC) and non-Hunner-type IC based on the presence of Hunner lesions, but there is consensus that these should be separated as distinct diseases [4,5]. In Japan, it is divided into IC and BPS based on the presence of Hunner lesions.

There is no established treatment for HIC, and it is designated as an intractable disease in Japan [6]. According to the symptoms, current treatment methods include dietary guidance, medication, and surgical procedure, and commonly used therapeutic agents include analgesic, antidepressant, antiallergic, and immunosuppressive agents [7]. In addition, bladder hydrodistension and bladder instillation treatments using heparin, steroids, or dimethyl sulfoxide (DMSO) were also performed [8,9,10].

In Japan, Kampo formulae are commonly used clinically in addition to Western medicines. Kampo formulae are herbal drugs used in traditional Japanese medicine that consist of several crude drugs, the extracts of which are generally administered. They have been used in many cases for the treatment of lower urinary tract conditions [11,12,13,14,15]. The Kampo formula Yokukansan (YKS) has analgesic properties and is used in various painful conditions [16,17,18,19,20]. It also demonstrated beneficial effects in patients with HIC; however, YKS has not yet been commonly used for the treatment of HIC, and its efficacy and the underlying mechanism of action have not been clarified. Since Kampo formulae generally exert antioxidant activity [21,22], and DMSO, used for bladder instillation treatment [8,9,10], also has an antioxidant effect [23,24], we hypothesized that the antioxidant activity of YKS might be underlying its efficacy in alleviating the symptoms of HIC.

Therefore, the purpose of the present study was to investigate the effects of YKS in HIC using an experimental animal model and to verify its antioxidant activity and its role as the possible underlying mechanism of action.

## 2. Materials and Methods

### 2.1. YKS

Dry powdered extracts of YKS (Lot No. 2110054010) used in the present study were supplied by Tsumura and Co. (Tokyo, Japan). The seven herbs (Table 1) were mixed and extracted with purified water at 95.1 °C for 1 h; the soluble extract was separated from the insoluble waste and concentrated by removing water under reduced pressure. The three-dimensional high-performance liquid chromatography profile chart of YKS showing the major chemical compounds was provided by Tsumura and Co. (Figure 1). The details of chemical compounds are shown in Table A1 of Appendix A. Kampo formulae are strictly stipulated by the Japanese pharmacopeia that the extract contains the specified active ingredient in a certain concentration, and its quality is maintained [25].

### 2.2. Antioxidant Capacity of YKS

Qualitative and quantitative analysis of hydroxyl radical (·OH) generated by the Fenton reaction was performed. In the Fenton reaction, ferrous iron reacts with hydrogen peroxide, which is reduced to generate ·OH. Since ·OH has a short life, it is detected using the electron spin resonance (ESR) spin trapping method. In the present study, 5,5-dimethyl-1-pyrroline-N-oxide (DMPO; LM-2110, LABOTEC Co., Ltd., Tokyo, Japan) was used as a trapping reagent, and ·OH was detected as a DMPO-OH radical.

In detail, after weighing dry powdered YKS (5, 10, 20, and 30 mg), 100 μL of 1 mM FeSO_4_ was added to each YKS sample and mixed. Ascorbic acid samples (0.67, 1, 2, and 2.5 mg) (012-04802, FUJIFILM Wako Chemicals, Osaka, Japan) were also prepared as comparative samples. An aliquot of water (120 μL) was mixed with 40 μL of each sample preprocessed with FeSO_4_—20 μL of 89 mM DMPO and 20 μL of 100 mM H_2_O_2_ in a plastic cuvette for 1 min. Next, the sample was transferred to a cuvette cell for ESR spectrometry, and the ESR spectrum was recorded on an ESR spectrometer (EMX-micro ESR spectrometer, Bruker BioSpin, Ettlingen, Germany). The measurement conditions for ESR were as follows: field sweep, 3450−3500 G; field modulation frequency, 9.84 GHz; field modulation width, 0.1 mT; sweep time, 30 s; time constant, 0.01 s; microwave frequency, 9.420 GHz; and microwave power, 10 mW. The ESR spectrum of chromium (Cr^3+^) held in the ESR cavity was used as an internal standard.

Measurements were performed twice for each sample, and the average values were used in the analysis. The ·OH scavenging capacity was shown by the ratio of the radical quantity generated when YKS or ascorbic acid was added ([DMPO-OH]_s_) to the radical quantity generated without YKS or ascorbic acid ([DMPO-OH]_0_).

### 2.3. Animal Investigations

#### 2.3.1. Animals

Animal experiments were performed using 8-week-old female Wistar rats (Nippon Bio-Supp. Center, Tokyo, Japan). Rats were housed three to four per cage (W 24 × L 40 × H 20 cm) under a 12 h light/dark cycle in our animal facility with a controlled environment (temperature 25 ± 2 °C and humidity 55 ± 5%). Food (CLEA Japan, CE-2, Tokyo, Japan) and water were provided ad libitum.

The experiments were performed in accordance with the guidelines of the Committee of Animal Care and Welfare of Showa University. All experimental procedures were approved by the Committee of Animal Care and Welfare of Showa University (certificate number: 02085, date of approval: 1 April 2020). An effort was made to minimize the number of animals used and their suffering.

#### 2.3.2. Groups and Induction of HIC

We used a toll-like receptor-7 (TLR7) agonist-induced rat model of HIC [26,27]. Similar to human HIC, this model exhibits frequent voiding and bladder pain-like behavior, and inflammatory cell infiltration, edema, and capillary congestion are observed in the suburothelial bladder layer.

Twenty-one rats were randomly divided into three groups, as follows: control, HIC, and YKS-treated HIC (YKS+HIC) groups. Since this model is an acute model and the symptoms do not last long [26], we decided to examine the effect of YKS by preadministration. YKS was mixed with powdered rodent chow (CE-2: CLEA Japan) at a concentration of 3% and fed to the YKS+HIC group for 7 days prior to the induction of cystitis. This dose was chosen based on the effective doses of YKS in our previous study [28,29]. Rats that were not treated with YKS were fed powdered chow only.

All rats were lightly anesthetized using intraperitoneally (i.p.) administered pentobarbital sodium (30 mg/kg; Somnopentyl, Kyoritsu Seiyaku Co., Tokyo, Japan). A polyethylene catheter (PE-10; Becton Dickinson, Franklin Lakes, NJ, USA) was transurethrally inserted into the bladder, and urine was drained. Next, 200 μL of 4.5 mM loxoribine (AG-CR1-3584, Adipogen Corp., San Diego, CA, USA), a selective TLR7 agonist, was instilled slowly in rats in the HIC and YKS+HIC groups. In the control group, distilled water was instilled instead of loxoribine. The catheter was kept in place for 60 min, after which the instilled liquid was drained.

#### 2.3.3. Von Frey Test

The von Frey test was performed to assess bladder pain at baseline, prior to HIC induction (day 1), and on days 2 and 3. Studies assessed rodent responses to von Frey filaments of varying diameters to determine the thresholds for noxious stimuli for the body part of interest [30,31]. The rats were placed in test cages with a wire mesh bottom for 30 min before the test. The surface of the lower abdomen close to the bladder was pushed, and the minimum value that caused withdrawal behavior was determined as the threshold.

#### 2.3.4. Measurement of Oxidative Stress

On day 3, the bladders of all rats were removed under deep anesthesia with pentobarbital sodium (50 mg/kg i.p.; Somnopentyl). A part of each bladder was plunged into 5 volumes (*v*/*w*) of a protein extraction reagent (N-PER, Thermo Scientific, Rockford, IL, USA) and then homogenized twice at 1500 rpm for 120 s using a tissue homogenizer (BMS-M10N21, Bio Medical Science Inc., Tokyo, Japan). The homogenate was centrifuged at 2000× *g* for 10 min, and the supernatant was collected. The total protein concentrations in the samples were determined using the Pierce™ BCA Protein Assay Kit (23227; Thermo Scientific, Waltham, MA, USA). The concentrations of all samples were standardized based on the amount of protein. Samples were subsequently stored at −80 °C until the analysis. The remaining part of each bladder was 4% formalin-fixed, embedded, and frozen in Tissue-Tek optimum cutting temperature (OTC) compound (Tissue-Tek OCT, Sakura Finetek, Torrance, CA, US) and stored at −80 °C until use.

The following oxidative stress markers were evaluated:OH—detected using the ESR spin trapping method, as described above. An aliquot of water (120 μL) was mixed with 20 μL of each bladder sample—20 μL of 1.0 M DMPO, 20 μL of 10 mM FeSO_4_, and 20 μL of 100 mM H_2_O_2_ in a plastic cuvette for 1 min. The subsequent measurements are as described above;Reactive oxygen metabolites (ROMs)—oxidative stress level was measured using the diacron reactive oxygen metabolites (d-ROMs) test (d-ROMs Kit; Diacron International S.r.l., Grosseto, Italy) [32]. ROMs of a biological sample, primarily hydroperoxides, are able to generate alkoxyl and peroxyl radicals, according to Fenton’s reaction. These radicals oxidize an alkyl-substituted aromatic amine (N,N-dietylparaphenylendiamine), thus producing a pink-colored derivative that is photometrically quantified at 505 nm using the free radical analytical system (FREE carpe diem; Diacron International S.r.l.) [33]. The concentration of ROMs is directly correlated with the color intensity and is expressed as Carratelli Units (1 CARR U = 0.08 mg hydrogen peroxide/dL) [34];8-hydroxy-2′-deoxyguanosine (8-OHdG)—8-OHdG produced by oxidative damage to DNA by reactive oxygen species (ROS) was detected using enzyme-linked immunosorbent assay and fluorescent immunostaining methods. The concentrations of 8-OHdG in the homogenate samples were measured using a kit (KOG-HS10E, Japan Institute for the Control of Aging, Fukuoka, Japan). The assay sensitivity was <0.125 ng/mL, and the intra- and inter-assay CVs were 2.1% and 7.1%, respectively. All measurement procedures were conducted according to the manufacturer’s instructions.

The frozen bladders in the OCT compound were cut into 25 µm sections using a cryostat (CM1860; Leica Biosystems, Nussloch, Germany). Sections were incubated overnight at 4 °C with rabbit anti-8-OHdG antibody (1:200, bs-1278R, Bioss, Woburn, MA, USA). Sections were then incubated for 2 h with fluorophore-tagged secondary antibody (donkey anti-rabbit Alexa Fluor 555, 1:1000, #A31572, Thermo Fisher Scientific, Waltham, MA, USA). Nuclei were counterstained with DAPI (4′,6-diamidino-2-phenylindole, 1:1000, Thermo Fisher Scientific). Samples were imaged using a confocal laser scanning fluorescence microscope (FV1000D, Olympus, Tokyo, Japan).

### 2.4. Statistical Analysis

All experimental data are expressed as medians (25% and 75% percentiles). All data analyses were performed using statistical software (SPSS 15.0J for Windows; SPSS Japan, Tokyo, Japan). Because the data had a skewed distribution, differences among the three groups were analyzed using the Kruskal–Wallis test. If there were significant differences in the Kruskal–Wallis test, the Bonferroni post hoc test was used to analyze the differences between each group. Statistical significance was set at *p* < 0.05.

## 3. Results

### 3.1. OH Scavenging Capacity of YKS

Compared to the same amount of ascorbic acid, a well-known antioxidant [35], YKS demonstrated a lower ·OH scavenging capacity (Figure 2a). Since YKS is a mixture and each has a different digestion and absorption process, we considered that these would be administered to humans and compared them at the appropriate daily dose. The daily dose of YKS for adults is 3.25 g [36], and the recommended dose of ascorbic acid for adults by the National Institutes of Health (office of dietary supplements) is 90 mg for men and 75 mg for women [37]. The daily dose of YKS is more than 30 times higher. With reference to these doses, ascorbic acid (0.67 mg) and YKS (20 mg), and ascorbic acid (1 mg) and YKS (30 mg) were compared. YKS shows higher (· OH) scavenging capacity (Figure 2b).

### 3.2. Animal Investigations

#### 3.2.1. Von Frey Test

Withdrawal thresholds were measured at baseline (day 1) and on days 2 and 3. Since the baseline values varied across animals, the values obtained on day 2 (Figure 3a) and day 3 (Figure 3b) are shown by the rate of change in relation to a baseline value of 1. On both days 2 and 3, a significant decrease in the threshold value was observed in the HIC group compared with the control group; however, the decrease was suppressed by the administration of YKS (day 2, *p* = 0.023; day 3, *p* = 0.072).

#### 3.2.2. Oxidative Stress Parameters

Representative stress parameters were measured to examine the changes in oxidative stress levels in the bladder wall (Table 2). The specific ESR signal (1:2:2:1) of the DMPO-OH adduct was detected (Figure 4a). The values were then expressed and converted to moles (Figure 4b). ·OH showed a significant increase in the HIC group compared with the control group; however, the increase was significantly suppressed by the administration of YKS.

ROMs and 8-OHdG also showed increasing tendencies in the HIC group; however, the increases were significantly suppressed in the YKS+HIC group (Figure 5a,b). In particular, the 8-OHdG expression in the HIC group increased in the bladder epithelium (Figure 5c–h).

## 4. Discussion

In the present study, the administration of YKS inhibited bladder pain-like behavior and suppressed the increase in expression of oxidative stress markers in the bladder wall in a rat model of HIC.

Kampo formulae were reported to have certain antioxidant activity [21,22]. In this study, the ·OH scavenging capacity of YKS was verified by comparing it with that of ascorbic acid, which is a well-known antioxidant agent. The result showed that ascorbic acid exhibited higher ·OH scavenging capacity than YKS. However, YKS exhibited higher ·OH scavenging capacity when adjusted for the daily dose. We will also compare these antioxidant effects in vivo. YKS contains many ingredients, and all its main ingredients, such as liquiritin [38], liquiritin apioside [39], and ferulic acid [40], have antioxidant capacity. In order to determine which ingredient played a central role in its antioxidant activity, we will further investigate the individual ·OH scavenging capacity of YKS ingredients.

Numerous animal studies suggested the involvement of ROS and the effect of antioxidants in IC. ROS played an important role in the cyclophosphamide (CP)-induced cystitis model, and antioxidants such as melatonin, alpha-tocopherol, ascorbic acid, and daidzin were found to ameliorate bladder damage induced by CP [41,42,43]. Furthermore, in CP- or ifosfamide-induced cystitis animal models, the metabolite acrolein caused bladder inflammation, which was prevented by ROS scavengers or antioxidants [43,44,45]. In clinical practice, bladder instillation treatment is performed using DMSO, which is a hydroxyl radical scavenger [8,9,10]. Therefore, we hypothesized that ROS plays an important role in the pathology of IC and that the antioxidant effect of YKS may contribute to IC suppression.

In the present study, to verify the effect of YKS on HIC, we used the TLR7 agonist-induced HIC animal model established by Ichihara et al. [26]. They found an increase in the TLR7 mRNA expression in the bladder of patients with HIC and established the HIC model by applying a TLR7 agonist, loxorbine. This model of rats showed bladder pain-like behavior, such as licking or biting the skin of the lower abdomen [26,27]. Bladder pain was assessed by the von Frey test, which is frequently used in cystitis model rats [30,31]. The withdrawal threshold in the HIC group was significantly lower than that in the control group following HIC induction; however, the decline was suppressed in the YKS+HIC group, which suggested the potential efficacy of YKS in alleviating HIC symptoms.

Next, we measured three oxidative stress parameters in the bladder wall. ·OH is the most reactive oxygen metabolite of various free radicals, and 8-OHdG is produced by oxidative damage of DNA induced by ·OH. The d-ROMs test evaluates the degree of oxidative stress by measuring the total peroxidative substances generated by various ROS, including ·OH. The level of ·OH increased significantly, and its reaction products 8-OHdG and ROMs also showed increasing tendencies; however, the increase was significantly suppressed by the administration of YKS. As mentioned above, ROS may play an important role in IC development [41,42,43,44,45,46], and the production of ·OH, which is one of ROS, increased in the TLR7 agonist-induced HIC model as well. Furthermore, the premedication with YKS was able to suppress ROS production.

TLR7 is expressed in immune cells, such as B cells, macrophages, and dendritic cells, recognizes single-stranded RNA of viruses and bacteria, and is involved in infection defense by inducing the production of interferon and inflammatory cytokines [47,48]. However, it was also reported to be involved in the development of autoimmune diseases, such as systemic lupus erythematosus, by erroneously reacting with own RNA [49]. In neutrophils and B cells, TLR7 agonists activate NADPH oxidase, increasing ROS production [50,51]. Influenza A virus infection increases oxidative stress by upregulating the expression of TLR7 [52]. Liquiritin [53], liquiritin apioside [54], and ferulic acid [55] included in YKS are reported to suppress the activity of NADPH oxidase. It is possible that these ingredients were involved in the YKS antioxidant activity exhibited in this study, and in the future, we will investigate the effect of YKS on TLR7 and NADPH oxidase in this experimental system.

To our knowledge, there has never been a study in which YKS was applied to HIC model animals. Our first goal of the present study was to investigate whether YKS had any effect on HIC, and the study was conducted without positive controls. Since the efficacy was suggested in the present study, we plan to further investigate the mechanism of action and examine whether it is useful in comparison with other therapeutic agents by setting positive controls. There are some more limitations to the present study. We only evaluated pain-like behavior and did not evaluate urination or the inflammatory findings of the bladder wall; thus, these analyses will be performed in the next study. It is necessary to clarify whether the inhibition of ·OH production is involved in the mechanism of action underlying the analgesic effect of YKS. Furthermore, the currently widely used chemical-induced IC model animals, including the HIC model we used, show only the symptoms of acute cystitis and cannot be evaluated for chronic cystitis. In the future, we will also use and verify a model with chronic symptoms [30].

## 5. Conclusions

In this study, we explored the effects of YKS in a rat model of HIC and found that it was effective in suppressing HIC symptoms. Furthermore, our findings indicated that the antioxidant activity of YKS is involved in the mechanism of action underlying its analgesic effect. These findings suggest that YKS may suppress the onset and/or exacerbation of HIC.

## Figures and Tables

**Figure 1 medicina-58-00810-f001:**
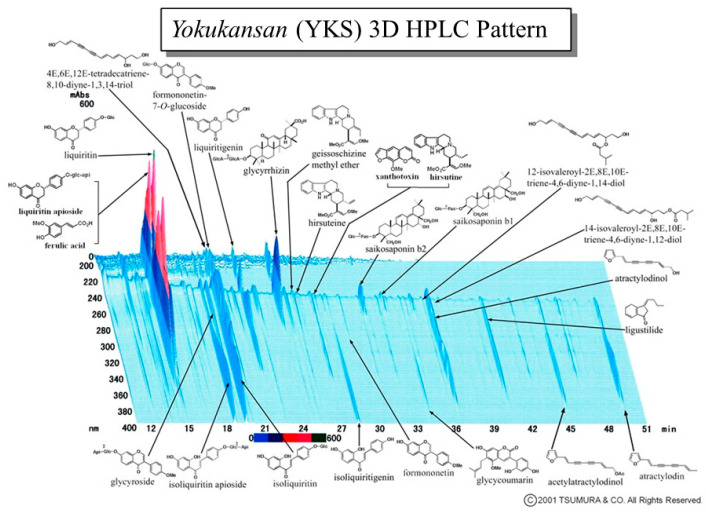
Three-dimensional high-performance liquid chromatography (3D-HPLC) profile chart of the major chemical compounds in Yokukansan (YKS).

**Figure 2 medicina-58-00810-f002:**
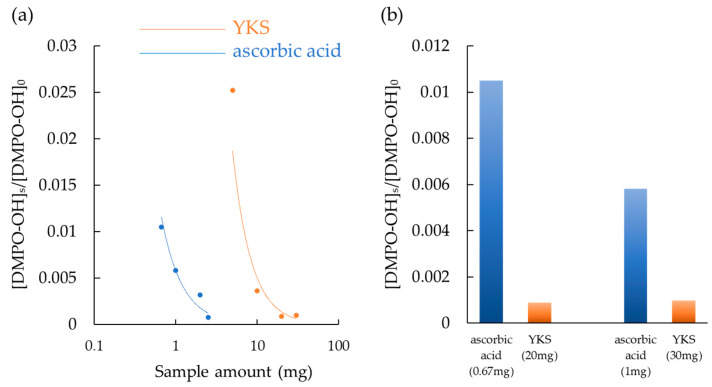
Hydroxyl radical (·OH) scavenging capacity. The ratio of the radical quantity generated when YKS or ascorbic acid was added ([DMPO-OH]_t_) to the radical quantity generated without YKS or ascorbic ([DMPO-OH]_0_). (**a**) Both showed higher ·OH scavenging capacity in a dose-dependent manner, with ascorbic acid higher. (**b**) YKS exhibited higher ·OH scavenging capacity when adjusted for the daily dose and compared. YKS, Yokukansan; DMPO, 5,5-dimethyl-1-pyrroline-N-oxide.

**Figure 3 medicina-58-00810-f003:**
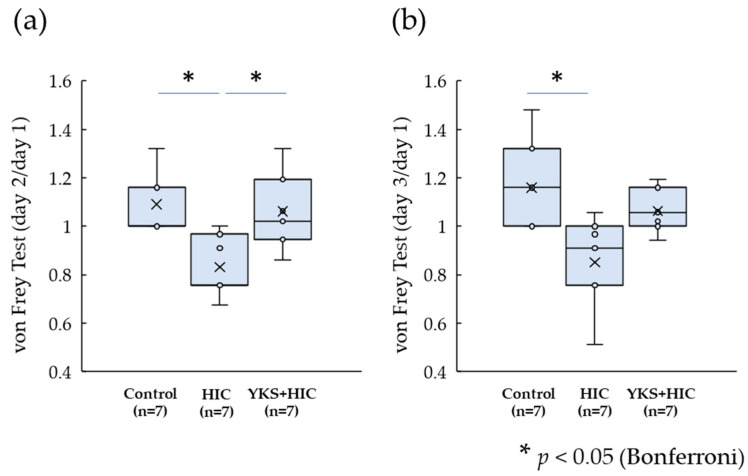
Von Frey Test. The values obtained on day 2 (**a**) and day 3 (**b**) are shown by the rate of change in relation to a baseline value of 1. Horizontal lines within boxes denote median values, and x marks denote the mean values. * Significant difference (*p* < 0.05) (Kruskal–Wallis test with post hoc test and Bonferroni correction). HIC, Hunner-type interstitial cystitis; YKS, Yokukansan.

**Figure 4 medicina-58-00810-f004:**
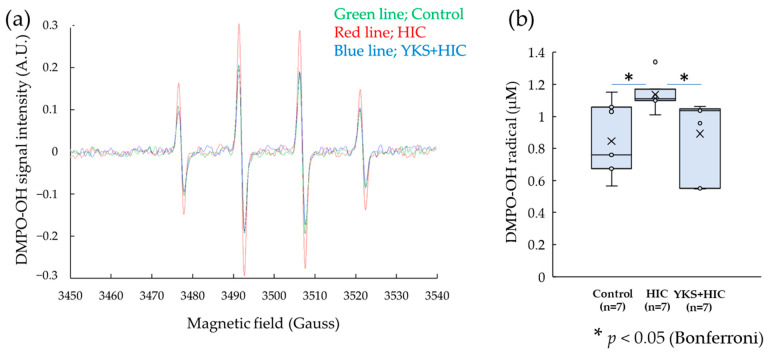
DMPO-OH radical. (**a**) DMPO-OH signals detected by ESR. (**b**) the intensity of the DMPO-OH signal. Horizontal lines within boxes denote median values, and × marks denote the mean values. ·OH significantly increased in the HIC group compared with the control group; however, the increase was significantly suppressed by the administration of YKS. * Significant difference (*p* < 0.05) (Kruskal–Wallis test with the post hoc test and Bonferroni correction). HIC, Hunner type interstitial cystitis; YKS, Yokukansan; DMPO, 5,5-dimethyl-1-pyrroline-N-oxide; ·OH; Hydroxyl radical.

**Figure 5 medicina-58-00810-f005:**
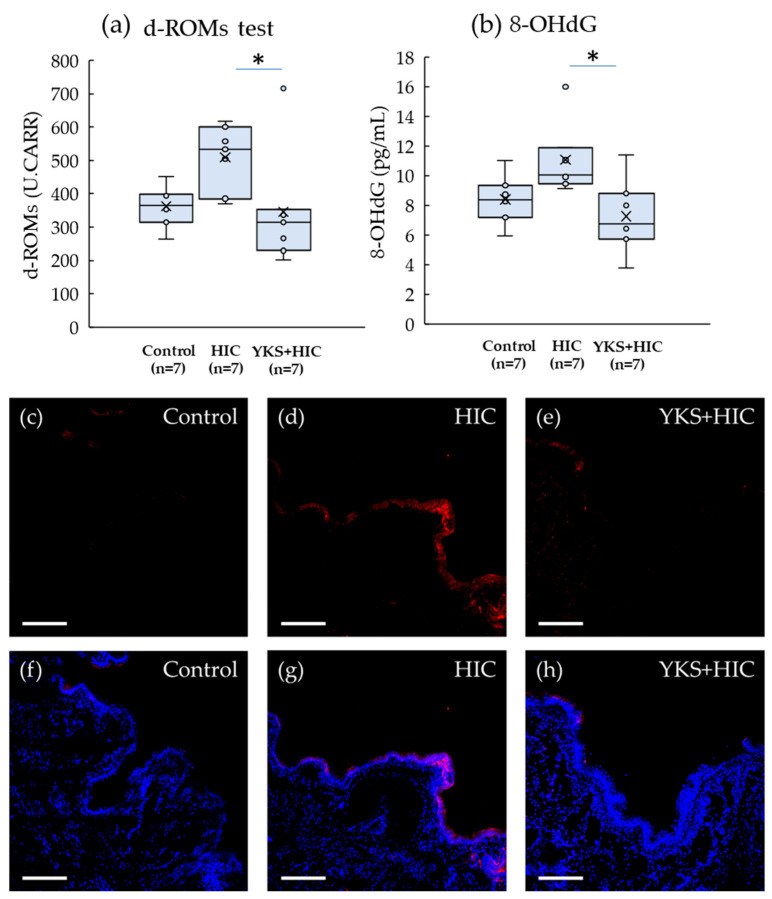
Oxidative Stress Parameters. (**a**) d-ROMs test; (**b**) 8-OHdG. Horizontal lines within boxes denote median values, and x marks denote the mean values. * Significant difference (*p* < 0.05) (Kruskal–Wallis test with the post hoc test and Bonferroni correction). (**c**–**h**) Immunofluorescent staining of 8-OHdG in the bladder. The upper row (**c**–**e**) includes only 8-OHdG, and the lower row (**f**–**h**) includes 8-OHdG and nuclei. Red, 8-OHdG; blue, DAPI (nuclei). The white scale bar is 200 μm. HIC, Hunner type interstitial cystitis; YKS, Yokukansan; d-ROMs, diacron reactive oxygen metabolites; 8-OHdG, 8-hydroxy-2′-deoxyguanosine; DAPI, 4′,6-diamidino-2-phenylindole.

**Table 1 medicina-58-00810-t001:** The component galenicals of Yokukansan (YKS).

Official Name (Upper Row) English Name (Lower Row)	Botanical Family	Amount
Rhizome of *Atractylodes japonica* Koidz. ex Kitam. *Atractylodes Lancea*	*Compositae*	4.0 g
Strain of *Pinus densiflora* Siebold and Zucc. *Poria*	*Polyporaceae*	4.0 g
Rhizome of *Cnidium officinale* Makino *Cnidium Rhizome*	*Umbelliferae*	3.0 g
Hook-bearing stems of *Uncaria rhynchophylla* (Miq.) Miq. *Uncaria Thorn*	*Rubiaceae*	3.0 g
Root of *Angelica acutiloba* (Siebold and Zucc.) Kitag. *Japanese Angelica*	*Umbelliferae*	3.0 g
Root of *Bupleurum falcatum* L. *Bupleurum*	*Umbelliferae*	2.0 g
Root of *Glycyrrhiza uralensis* Fisch. *Glycyrrhiza*	*Laguminosae*	1.5 g

Weights indicate the daily dose, and these are mixed and extracted with 600 mL of purified water (at 95.1 °C).

**Table 2 medicina-58-00810-t002:** Oxidative Stress Parameters.

	Control (*n* = 7)	HIC (*n* = 7)	YKS+HIC (*n* = 7)	Kruskal–Wallis Test (*p*-Value)
DMPO-OH radical (μM)	0.76 (0.67, 1.04)	1.11 (1.10, 1.15)	1.04 (0.75, 1.04)	0.018 *
d-ROMs (U.CARR)	364.15 (333.50, 395.93)	533.31 (444.64, 579.40)	314.80 (247.77, 343.79)	0.019 *
8-OhdG (pg/mL)	8.38 (7.71, 9.04)	10.06 (9.70, 11.47)	6.74 (6.08, 8.40)	0.011 *

All values are expressed as median (25% and 75% percentiles). * *p* < 0.05 (Kruskal–Wallis test). HIC, Hunner type interstitial cystitis; YKS, Yokukansan; d-ROMs, diacron reactive oxygen metabolites; DMPO, 5,5-dimethyl-1-pyrroline-N-oxide; ·OH; Hydroxyl radical; 8-OHdG, 8-hydroxy-2′-deoxyguanosine.

## Data Availability

The data presented in this study are available from the corresponding author on reasonable request.

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
