# Peer review of "The Japanese Herbal Medicine Yokukansan Exerted Antioxidant and Analgesic Effects in an Experimental Rat Model of Hunner-Type Interstitial Cystitis"

_medicina, 2022, doi:10.3390/medicina58060810_

Round 1

Reviewer 1 Report

The manuscript under the title "The Japanese Herbal Medicine Yokukansan Exerted Antioxidant and Analgesic Effects in an Experimental Rat Model of Hunner-type Interstitial Cystitis" can be accepted after addressing the following comments:

1- An abbreviations section should be added as the manuscript include many abbreviations, to ease reading.

2- The major chemical compounds in Yokukansan (YKS) presented in Figure (1) are unclear and small better to be presented in a table including the name, structure, and chemical class of each compound. 

3- why a positive control wasn't used in this in vivo study to ease the comparison of the activity of the tested samples.

4- what is the significance of this study to measure the activity of an herbal preparation available in the market already used in therapy?

Author Response

Reviewer 1

Comments and Suggestions for Authors

The manuscript under the title "The Japanese Herbal Medicine Yokukansan Exerted Antioxidant and Analgesic Effects in an Experimental Rat Model of Hunner-type Interstitial Cystitis" can be accepted after addressing the following comments:

Thank you for your polite review while you are busy.

Point 1: An abbreviations section should be added as the manuscript include many abbreviations, to ease reading.

Response 1: Thank you for your precise opinion. We have added an abbreviations section (L358).

Point 2: The major chemical compounds in Yokukansan (YKS) presented in Figure (1) are unclear and small better to be presented in a table including the name, structure, and chemical class of each compound.

Response 2: We added another table as appendix because it is too large to show in the text (L370).

Point 3: why a positive control wasn't used in this in vivo study to ease the comparison of the activity of the tested samples.

Response 3: We thought that what you pointed out was quite right. To our knowledge, there has never been a study in which YKS was applied to HIC model animals, and our first goal of this study was to investigate whether YKS had any effect on HIC. Furthermore, in order to minimize the number of animals used, this study was conducted without setting positive controls. Since the efficacy was suggested in the present study, we plan to further investigate the mechanism of action and to examine whether it is useful in comparison with other therapeutic agents by setting a positive control. We have added this to the limitation (L320).

Point 4: what is the significance of this study to measure the activity of an herbal preparation available in the market already used in therapy?

Response 4: We have experienced effective cases in some patients, but YKS has not yet been commonly used for HIC(L53). For the drug repositioning of YKS, we wanted to clarify the efficacy and mechanism of action against HIC.

Reviewer 2 Report

The manuscript studied the antioxidant capacity of YKS in a Hunner-type interstitial cystitis animal model. Such a finding is important to understand the potential mechanism of action of YSK in HIC treatment. At the same time, the authors understand the limitations of current studies and laid out future study plans. The manuscript can be accepted after followed modifications are addressed. 

1.  Additional clinically relevant positive control (such as DMSO mentioned in the manuscript) should be used in the animal model. 

2. A relative concentration of all major compounds in YKS is helpful to elucidate active ingredient(s) in YKS extracts (Figure 1) and/or constituent(s) among the seven YKS components (Table 1).  

3. Herbal medicine may have different concentrations of active ingredients due to variations from planting and harvest. Although it is out of the scope of current work, the authors should comment on this issue.

Author Response

Reviewer 2

Comments and Suggestions for Authors

The manuscript studied the antioxidant capacity of YKS in a Hunner-type interstitial cystitis animal model. Such a finding is important to understand the potential mechanism of action of YSK in HIC treatment. At the same time, the authors understand the limitations of current studies and laid out future study plans. The manuscript can be accepted after followed modifications are addressed.

Thank you for your polite review while you are busy.

Point 1: Additional clinically relevant positive control (such as DMSO mentioned in the manuscript) should be used in the animal model.

Response 1: We thought that what you pointed out was quite right. To our knowledge, there has never been a study in which YKS was applied to HIC model animals, and our first goal of this study was to investigate whether YKS had any effect on HIC. Furthermore, in order to minimize the number of animals used, this study was conducted without setting positive controls. Since the efficacy was suggested in the present study, we plan to further investigate the mechanism of action and to examine whether it is useful in comparison with other therapeutic agents by setting a positive control.(L320)

Point 2: A relative concentration of all major compounds in YKS is helpful to elucidate active ingredient(s) in YKS extracts (Figure 1) and/or constituent(s) among the seven YKS components (Table 1). 

Point 3: Herbal medicine may have different concentrations of active ingredients due to variations from planting and harvest. Although it is out of the scope of current work, the authors should comment on this issue.

Response 2&3: Japanese Kampo formulae are strictly stipulated by the Japanese pharmacopoeia that the extract contains the specified active ingredient in a certain concentration or higher, and those that do not meet the standards are not accepted as drugs. Therefore, even if the production areas are different, a certain level of quality is maintained.(L71)

Thus, although the amount of the specified components have been measured, so far we have not made a rigorous quantification of the other components. In the future, we will quantify the main components in order to identify which components are importantly acting.

Round 2

Reviewer 1 Report

The authors adequately fulfilled all the requirements mentioned in the first revision.

the manuscript can be accepted in its current form